# The Influence of Socio-Demographic Factors on Preference and Park Usage in Guangzhou, China

Yueshan Ma *, Paul Brindley  and Eckart Lange 

Department of Landscape Architecture, The University of Sheffield, Sheffield S10 2TN, UK;
p.brindley@sheffield.ac.uk (P.B.); e.lange@sheffield.ac.uk (E.L.)
* Correspondence: yma51@sheffield.ac.uk; Tel.: +86-18538841195

**Abstract:** Urban green space (UGS) provides a range of services to visitors and is particularly important for recreation and well-being. There are a number of approaches to research visitor accessibility, but implications for demographic differences of users are typically ignored. In order to more precisely model usage of UGS regarding visitor preference, this study used Guangzhou (China) as a case study, concentrating on residents' visitation to parks and their factors across different groups (for example, by gender, education level, age and visiting frequency). Online questionnaires from 2360 adults were collected on visiting preferences, such as traveling time, visiting frequency, visit duration within parks, visiting reasons and barriers. Results indicate that women were less likely than men to undertake longer walking trips to access parks (over 40 min). Elderly people tended to have longer visit durations, and lower-educated people tended to have shorter visiting times (particularly less than 15 min) in parks. Visit duration in parks had a positive association with walking time and a negative association with visiting frequency. Furthermore, the proportion of people visiting parks to relieve stress declined with increasing age. Infrequent park users (visiting parks less than once a month) rarely visited to gain inspiration or to socialize with strangers. Barriers to use of parks were correlated with socio-demographic factors, highlighting that older people identified poor quality of parks and long walking times as critical barriers. This study provides evidence that there is no one-size-fits-all modeling approach for UGS usage; instead, it demonstrates the importance of considering the socio-demographic characteristics of users.

**Keywords:** park usage; demographic difference; visiting preference; structured questionnaire survey; statistical analysis

## 1. Introduction

Urbanization continues to increase, with currently 55% of the world's population living in urban areas [1]. This has significant implications for the natural environment around the world [2,3]. The role of urban green space (UGS) in contributing to both the physical and mental health of humans (for example, helping with chronic disease care [4], reducing the risk of obesity [5–7], alleviating negative emotion [8], reducing physical disease and mental depression [9,10] and so on), is commonly emphasized and demonstrated by a growing body of research. Urban parks are a significant element of wider UGS with inherent benefits similar to those of UGS for the physical and psychological well-being of humans [9,11,12]. Therefore, visiting parks can be instrumental in influencing human well-being and health.

Most studies have explored factors affecting park visitation in terms of motivations. Attributes of UGS, such as quietness, space, facilities and so forth, are found to influence UGS use [13,14]. Recent studies have further identified that four attributes, namely socio-demographic features, spatial distribution of residents, individual characteristics and park features, conjointly lead to different modes of utilizing parks [15,16]. For example, the transportation mode is identified as being related to park use [15–18]. Reasons for visiting parks mainly refer to relaxation, exercise, socialization and companionship with

children [15,16,19–22]. At the same time, patterns of visiting parks are affected by different population groups of users, such as age [15,16], income level [23,24], education [25] and gender [26]. Although many studies have explored factors affecting park visits, they were mostly conducted based on a few selected parks; thus, they do not provide a comprehensive picture of park usage at a city scale.

China is a highly urbanized country encountering a serious loss of natural environments, which has caused uneven access to UGS within cities [27]. An exhaustive understanding of the factors influencing park usage could offer planners scientific foundations with which to improve the efficiency of the provision of green space. This is especially true considering the challenge of exploring areas for new UGSs in a heavily urbanized city considering the high cost of land and limited land resources [28]. Guangzhou, as the largest city in southern China [29], is pursuing an ambitious plan for providing new UGSs. According to the special planning of park construction and protection in Guangzhou (2017–2035) [30], 500 city street gardens and 27 new parks are to be constructed by 2025 in Guangzhou. Therefore, a better understanding of visitations to parks by local residents is needed to provide planners and local authorities with rational criteria for the planning of new parks and to improve the visiting experience and efficiency of parks. Shan [31] conducted an analysis on green space use in Guangzhou, stressing that walking was the most common transport mode, encouraging people to have frequent visitations to UGSs. Similarly, Zhan et al. [32] identified that good access to parks within walking distance attracted users due to daily exercise habits. Additionally, the attractiveness of parks could be reflected by an individual's visiting behavior [33], influencing park use [32,34].

Defining UGS can be problematic and can vary within different national contexts [35]. In China, UGSs are classified into five main types, including park green space, green buffer, square green space, attached green space and regional green space, depending on the nature of the land, proportion of green space, location and facilities. Detailed classification of parks in previous studies on UGS use could be determined using the size of parks [36,37], the geographic location of parks [38] and the facilities in parks [37], for example. To the authors' knowledge, there is no literature review exploring how socio-demographic factors affect use and visitation of green space typologies within a global context.

While there is a growing body of research examining the diverse factors influencing use of UGS and visiting preferences, this work was largely based on an analysis of a small number of selected parks rather than a comprehensive analysis across all parks of a city. As such, knowledge on the variations in park use and visiting preferences of different visitor groups remains understudied. Furthermore, existing research primarily identified the effect of transport modes on park visitation, identifying walking as the most important mode influencing users' visiting behavior. Therefore, a separate investigation on how preferences of and park usage by pedestrians across different population groups vary needs to be undertaken to inform planning and improvement schemes aiming to meet the demand for parks within walking distance.

Considering all parks in Guangzhou, this research examined the impact of socio-demographic factors (including age, gender, education and traveling time) on park usage, visiting frequency and visit duration. Additionally, motivations and barriers were explored for different visitor groups. This study aimed to offer insight into the following questions: (1) What is the socio-demographic composition of residents in Guangzhou visiting parks, and what are their preferences when walking to the parks? (2) Do the preferences for visiting parks vary across population groups? How do they affect park usage? (3) What are the motivations and barriers in terms of walking to visit parks, and do they vary for groups with diverse preferences and park uses?

## 2. Methodology

### 2.1. Study Area

This study focused on the entire area of Guangzhou. Guangzhou is the capital of the Guangdong province, in southern China (Figure 1a), and is located in the central zone of

the Pearl River Delta (Figure 1b) [39]. It has a subtropical oceanic monsoon climate. The Guangzhou metropolitan area has a good reputation for its large UGSs, and according to 2017 statistics, has a forest coverage rate of 42% and 41% green coverage for urban areas [29,40]. Nonetheless, with increasing urbanization, Guangzhou faces challenges such as the loss of green space and efforts to develop into a more sustainable city [41].

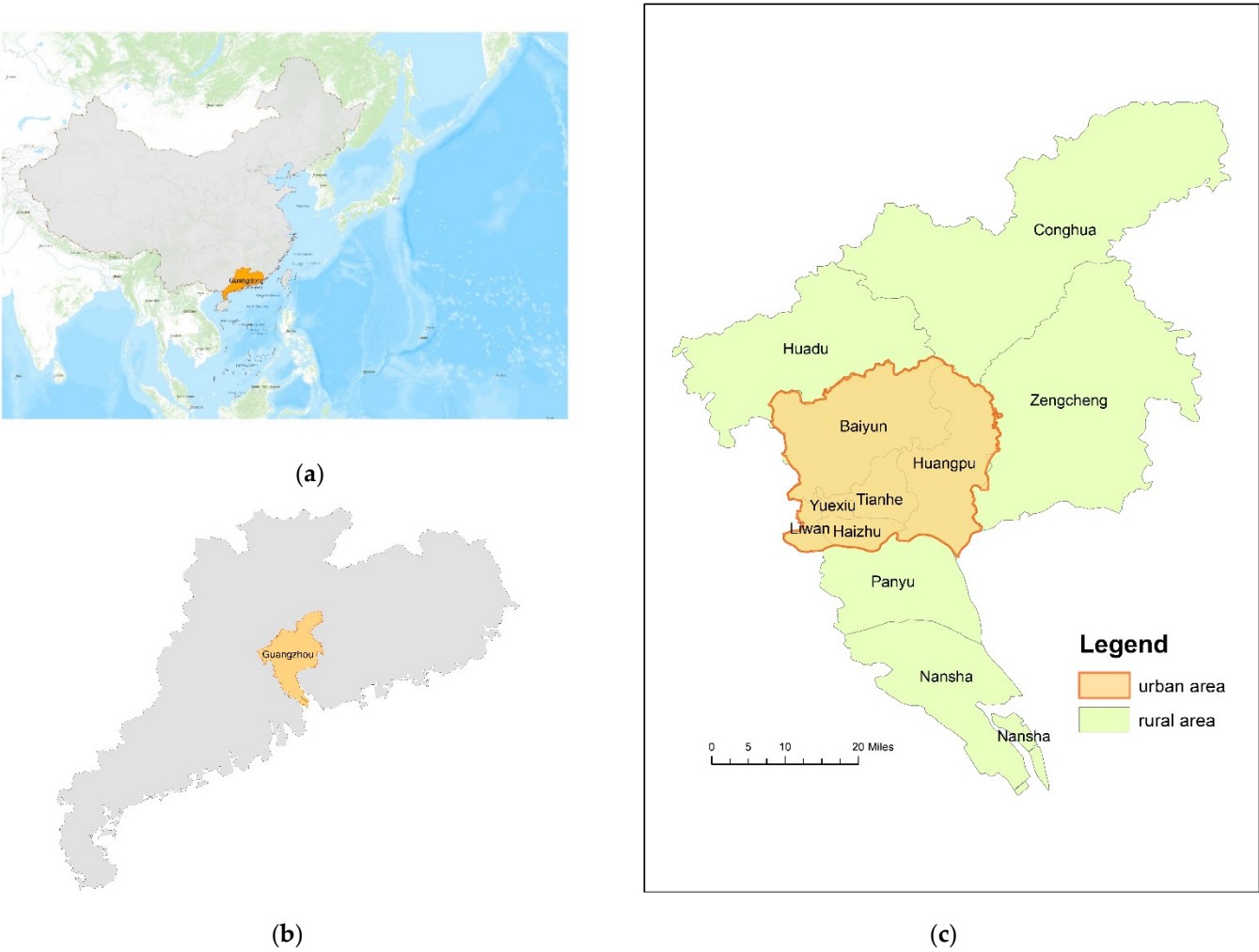

(**a**)

(**b**)

(**c**)

**Figure 1.** Location of study area, Guangzhou and administrative district division. (**a**) Location of Guangdong province in China; (**b**) location of Guangzhou in Guangdong; (**c**) spatial administrative areas of Guangzhou.

Guangzhou has eleven districts including 170 townships, within which the urban areas consist of six districts in the center. The rural area is constituted by another five districts (Figure 1c). In total, there are 494 parks in Guangzhou, and their entrance distribution is shown by green dots in Figure 2. As shown by the population density information within Figure 2, Guangzhou is characterized as a highly populated region, especially in urban areas [39].

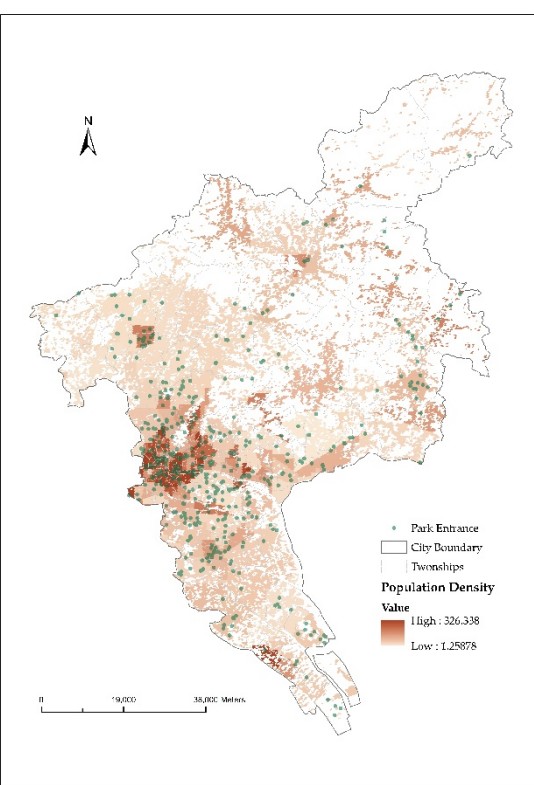

**Figure 2.** Spatial distribution of all park entrances with green dots and spatial distribution of population density (number of people per pixel; pixel size: 250 m × 250 m) with gradient red spots (the darker the color, the denser it is) in Guangzhou.

### 2.2. Data Collection and Measures

In this study, a survey was conducted to collect park usage data from residents in Guangzhou. It builds on a small number of studies in this area [16,19,22,23,31]. Questions were mainly closed-ended with predefined options. Three sub-topics of the questionnaire focused on preferences for urban parks, preferences for park usage by foot and the experience of walking to visit parks. Uniquely, instead of focusing on predetermined selected parks or the closest parks as in previous studies, the focus of the questions was on respondents' self-identification of the parks that they most frequently visited (first and second most-visited parks) as well as the park most visited by foot.

A pilot study was conducted to examine the logic and possible bias of questions [42]. The pilot (10 respondents) assisted in reducing ambiguity and the length of survey. A random sampling method achieved 2360 questionnaire respondents via popular social media in China including Weibo, WeChat, Zhihu, Baidu Tieba and Douban Group. Respondents needed to be over 18 and have lived in Guangzhou for three months or more. In total, there were 2343 fully completed questionnaires.

Measures of experience and attitudes, for example, ease of access, overall visiting experience, general park quality, etc., were conducted using a five-point Likert scale ranging from "Very difficult"/"Very unsatisfied" (coded 1) to "Very easy"/"Very satisfied (coded 5). Preferences for visiting parks consisted of transportation modes, traveling time, visiting frequency and visit duration within parks. Socio-demographic factors consisted of age, gender and education.

Preprocessing and analysis of the obtained questionnaire data were undertaken in Excel and IBM SPSS Statistics 26. Some descriptive data were quantified into dimensionless data with ordinal measurement (Table 1). A chi-square test was conducted to monitor the validation of sample data compared with the population census of Guangzhou [43]. Table 2 provides details on the demographic comparison between the census and respondent population, with corresponding weights computed for each socio-demographic group.

These weights were applied for subsequent analysis. Nonparametric tests investigated correlation between factors and park use conditions across visitor groups. The influence of crossover among factors on visiting preferences and park use was explored using regression models and cross tabulation, which were also used for exploring the differences in motivations for and barriers to park visitation across diverse visitor groups. To be specific, nonparametric tests (chi-square analysis) were used to investigate correlation between factors and park use conditions across visitor groups. The influence of crossover among factors on visiting preferences and park use was explored using linear regression models and cross tabulation, which were also used for exploring the differences in motivations for and barriers to park visitation across diverse visitor groups. Full details of the statistical tests undertaken can be found in Online Supplementary Table S1.

**Table 1.** Variable definition and descriptive statistics.

| Factors | Variables | Definition Value | Mean | St.d | Median |
|---|---|---|---|---|---|
| Walking time | Less than 15 min | 4 | 4.72 | 0.81 | 5.00 |
| | 16 to 30 min | 5 | | | |
| | 31 to 40 min | 6 | | | |
| | More than 40 min | 7 | | | |
| Visit duration | Less than 15 min | 3 | 4.84 | 0.96 | 5.00 |
| | 16 to 30 min | 4 | | | |
| | 31 to 45 min | 5 | | | |
| | 46 to 75 min | 6 | | | |
| | More than 75 min | 7 | | | |
| Visiting frequency | Less than once a month | 3 | 6.57 | 1.11 | 7.00 |
| | Once a month | 4 | | | |
| | Twice a month | 5 | | | |
| | Once a week | 6 | | | |
| | Twice to three times a week | 7 | | | |
| | Every day or more often | 8 | | | |

**Table 2.** Validation of the representatives of population samples compared to census data and weights for each population group.

| Item | Category | Survey % | Observed Count | Census % | Expected Count | Difference | Weight | Chi-Square |
|---|---|---|---|---|---|---|---|---|
| Age | 18–24 | 11.99 | 283 | 18.13 | 1,862,543 | −6.14 | 1.51 | |
| | 25–30 | 35.97 | 849 | 14.29 | 1,468,606 | 21.68 | 0.40 | |
| | 31–40 | 50.21 | 949 | 24.33 | 2,499,930 | 25.88 | 0.48 | 7030.874 *** |
| | 41–50 | 9.66 | 228 | 19.48 | 2,001,680 | −9.82 | 2.02 | |
| | 51 and over | 2.16 | 51 | 23.77 | 2,442,389 | −21.61 | 11.00 | |
| Gender | Male | 61.40 | 1449 | 51.88 | 5,331,127 | 9.52 | 0.84 | 85.560 *** |
| | Female | 38.60 | 911 | 48.12 | 4,944,021 | −9.52 | 1.25 | |
| Education | Primary and below | 1.36 | 32 | 17.70 | 2,142,853 | −16.34 | 13.01 | |
| | Junior high school/High school | 9.92 | 234 | 61.71 | 7,469,316 | −51.79 | 6.22 | |
| | College | 50.08 | 1182 | 10.09 | 1,221,764 | 39.99 | 0.20 | 1649.74 *** |
| | Undergraduate | 31.48 | 743 | 9.23 | 1,116,627 | 22.25 | 0.29 | |
| | Postgraduate and above | 7.16 | 169 | 1.27 | 153,297 | 5.89 | 0.18 | |

*** indicates the asymptotic significance value is near to 0.

## 3. Results

### 3.1. Visiting Preferences across Demographic Groups for Parks Visited on Foot

Data analyzed in this section consist of respondent visits on foot.

Visiting preferences were found to be affected by gender characteristics, including walking time, visit duration within parks and visiting frequency (see Table 3).

**Table 3.** Differences in effects of socio-demographic characteristics on visiting preferences.

| Socio-Demographic Characteristics | Visiting Preferences for Their Most Frequently Visited Parks on Foot | Values | Sig. |
|---|---|---|---|
| Gender | Walking time | $X^2(3) = 7.852$ a | 0.049 * |
| | Visit duration | $X^2(4) = 1.023$ a | 0.906 |
| | Visiting frequency | $X^2(5) = 12.124$ a | 0.033 * |
| Age | Walking time | $X^2(12) = 109.673$ a | 0.000 *** |
| | Visit duration | $X^2(16) = 150.295$ a | 0.000 *** |
| | Visiting frequency | $X^2(20) = 139.513$ a | 0.000 *** |
| Education | Walking time | $X^2(6) = 50.076$ b | 0.000 *** |
| | Visit duration | $X^2(8) = 236.100$ b | 0.000 ** |
| | Visiting frequency | $X^2(10) = 183.991$ b | 0.000 *** |

"a" indicates 0 cells (0.0%) have an expected count of less than 5. "b" indicates more than 10.0% of cells have an expected count of less than 5. * Significant at $p < 0.05$; ** significant at $p < 0.01$; *** significant at $p < 0.001$.

An apparent difference was found in the model for tests of between-subjects effects. There was a statistically significant influence of the interrelation of walking time and gender on visiting frequency ($p < 0.050$). Figure 3a depicts the estimated mean value of visiting frequency for the most frequently visited park in different gender groups in terms of the increasing walking time. It highlights that males (as opposed to females) were more likely to undertake longer walking trips (over 40 min) to parks.

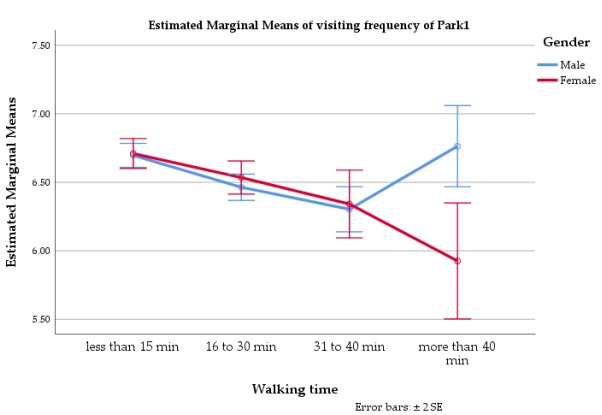

(**a**) Estimated mean value of visiting frequency for the most frequently visited parks with the effect of walking time and gender

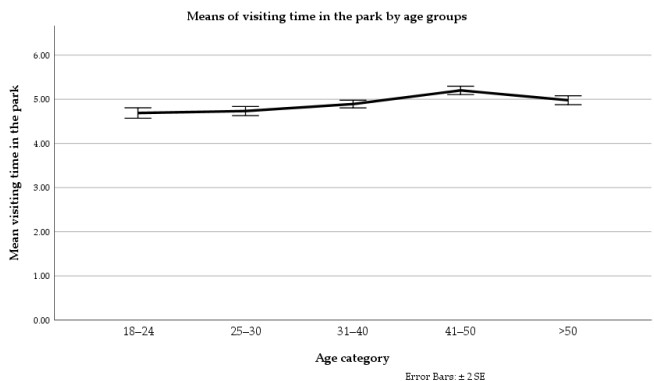

(**b**) Curve of means for visit duration within the most frequently visited parks across age groups

**Figure 3.** Line charts of correlation between visiting preferences with gender and age: (**a**) relationships between visiting frequency, walking time and gender; (**b**) relationship between visit duration and age.

The strong correlation between age groups and visiting preferences highlighted differences in visit duration across age populations. As displayed in Figure 3b above, older people tended to stay longer in parks than younger people. There was a slight decrease in visiting time for people aged over 50.

Table 4 illustrates the statistical significance of differences in visiting time categories across a subset of educated level groups. The "High school and below" group was significantly ($p < 0.05$) different from the other two higher-educated groups regarding having a visit duration of "Less than 15 min". This indicates that lower-educated people tended to have shorter visiting times, particularly less than 15 min, in parks.

**Table 4.** Comparison of statistics for visit duration among different education groups and results of z-test, suggesting the differences.

| Visit Duration | Statistic Index | Educated Groups | | | Total |
|---|---|---|---|---|---|
| | | High School and below | High School to College | Undergraduate and above | |
| Less than 15 min | Count | 298 [a] | 10 [b] | 12 [b] | 320 |
| | % within Education level | 17.5% | 4.4% | 5.4% | 14.9% |
| 16 to 30 min | Count | 444 [a] | 78 [b] | 68 [a,b] | 590 |
| | % within Education level | 26.1% | 34.7% | 30.8% | 27.5% |
| 31 to 45 min | Count | 538 [a] | 88 [a] | 84 [a] | 710 |
| | % within Education level | 31.6% | 39.1% | 38.0% | 33.1% |
| 46 to 75 min | Count | 314 [a] | 38 [a] | 45 [a] | 397 |
| | % within Education level | 18.4% | 16.9% | 20.4% | 18.5% |
| More than 75 min | Count | 108 [a] | 11 [a] | 12 [a] | 131 |
| | % within Education level | 6.3% | 4.9% | 5.4% | 6.1% |
| Total | Count | 1702 | 225 | 221 | 2148 |
| | % within Education level | 100.0% | 100.0% | 100.0% | 100.0% |

Each subscript letter denotes a subset of education level categories whose column proportions do not differ significantly from each other at the 0.05 level.

### 3.2. Effect of Factors on Park Usage

The transportation mode for visiting the most frequently visited park was closely related to traveling time ($X^2(9) = 187.953$, $p < 0.001$), access ease ($X^2(12) = 107.978$, $p < 0.001$) and their intercorrelation ($X^2(9) = 14.633$, $p < 0.01$). Walking accounted for an apparent higher proportion of all transportation modes for trips to parks requiring less than 15 min of traveling time (Figure 4a). When stating the corresponding access ease of each trip to their most frequently visited park, residents usually identified walking as the easiest mode even if traveling time varied (Figure 4b). In summary, walking can be considered the easiest and most frequent transportation method for visiting parks.

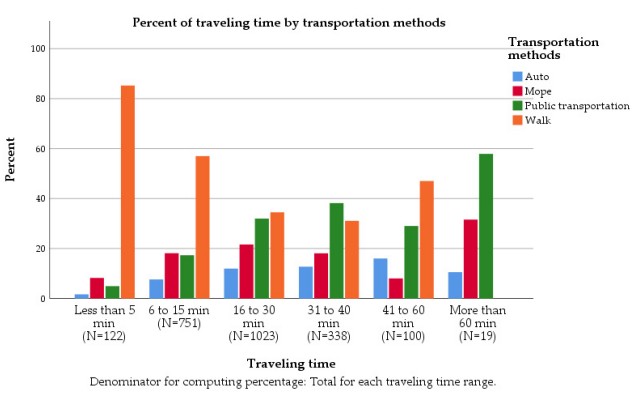

**(a)** The cluster bar variations in transportation modes for different traveling times to their most frequently visited park

**(b)** Estimated line charts of ease of access mean scores on time spent on traveling to corresponding most frequently visited parks across transportation methods

**Figure 4.** Differences in traveling time and ease of access for respondents across selections on transportation modes for each most frequently visited park: (**a**) summary of transportation modes and traveling time; (**b**) relationships between ease of access, traveling time and transportation modes.

The relationship between visit duration (for those visiting on foot) was explored against visiting frequency and the walking time required to visit the park. Firstly, visiting frequency significantly correlated with visit duration in the most frequently visited park on foot with a correlation value of $X^2$ (16) equal to 310.862 ($p < 0.001$). Secondly, walking time was strongly related to visit duration with a correlation value of $X^2$ (12) equal to 343.519 ($p < 0.001$). Specifically, Figure 5a,b show that there was a negative association

between the frequency of visits and visiting time (with more frequent visits associated with shorter visiting times), and a positive association between the walking time and time spent visiting parks (with average longer visiting times associated with longer walking distances), respectively.

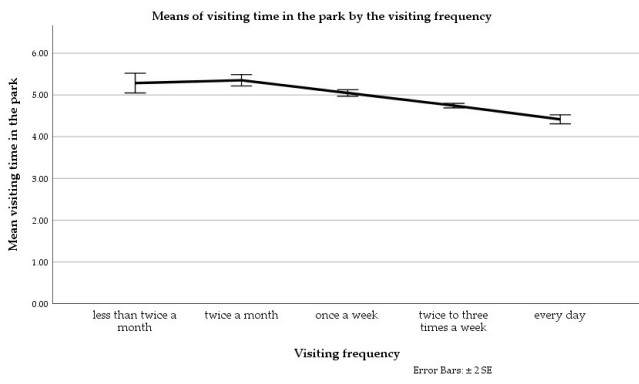 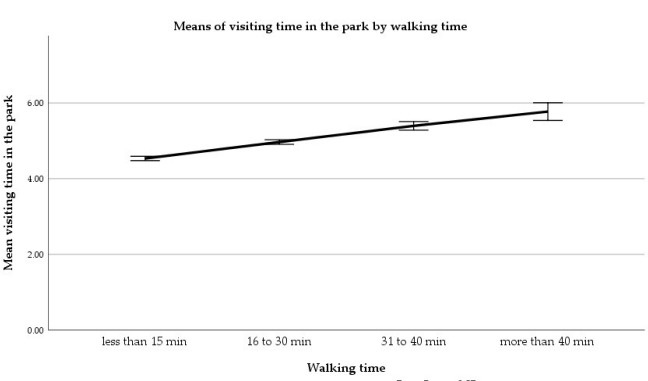

(**a**) Line chart of mean visiting time by visiting frequency in the most frequently visited park on foot

(**b**) Line chart of mean visiting time by walking time in the most frequently visited parks

**Figure 5.** Variations in mean visit duration in the individual's most frequently visited park on foot by (**a**) visiting frequency and (**b**) walking time.

Further exploration of the interaction effects (the combination of these two factors) identified a significant interrelation with park usage (visit duration). Analysis using a Univariate General Linear Model (UGLM) found that walking time and visiting frequency were significantly correlated with visit duration ($p < 0.001$, $R^2 = 0.198$), and the between-subjects effect of these two factors was also significantly correlated with visit duration ($p < 0.05$). Figure 6 shows the differences in mean visit duration by trips with different traveling times to parks as visiting frequency changed. This figure not only demonstrates the correlations between a single factor and visit duration as argued above, but also illustrates that more frequent and shorter trips to parks tended to correspond to shorter visiting times within parks.

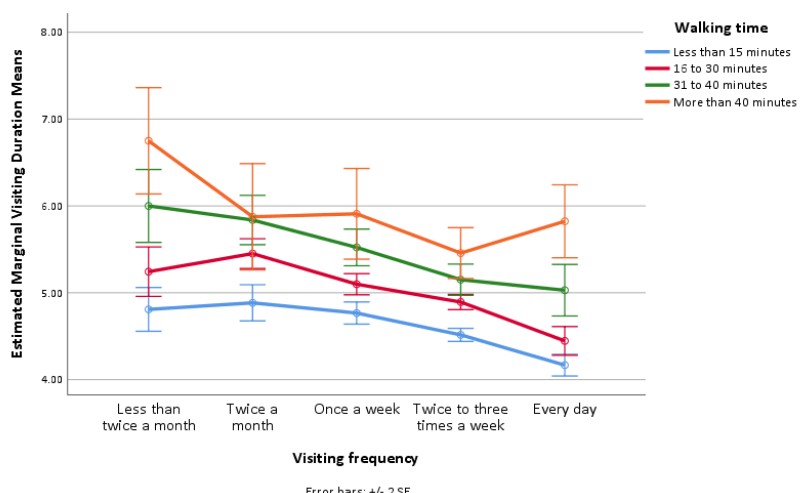

**Figure 6.** Estimated line charts of means for visiting time by the effect of walking time and visiting frequency.

### 3.3. Motivations for and Barriers to Park Visitation among Different Visitor Groups

Differences in motivations and barriers identified by respondents were found to correlate with visitor groups (age, gender, education and visiting frequency) and visiting preference elements (traveling time and visit duration) using chi-square analysis (Table 5).

**Table 5.** Correlating significance values of reasons for and barriers to visiting parks.

| | Reasons for Visiting Parks (N = 2263) | | | | | | |
|---|---|---|---|---|---|---|---|
| **Visitor Groups** | **Walking/Exercise** | **Relieve Stress** | **Gain Inspiration** | **Socialize with Strangers** | **Enjoy Views** | **Meet with Friends or Family** | **Passing by during Commute** |
| Age | 0.000 *** | 0.000 *** | 0.000 *** | 0.005 ** | 0.000 *** | 0.000 *** | 0.936 |
| Gender | 0.124 | 0.005 ** | 0.010 * | 0.000 *** | 0.000 *** | 0.690 | 0.406 |
| Education | 0.186 | 0.000 *** | 0.006 ** | 0.558 | 0.000 *** | 0.884 | 0.129 |
| Frequency | 0.086 | 0.432 | 0.000 *** | 0.000 *** | 0.09 | 0.767 | 0.712 |
| Visit preferences | | | | | | | |
| Traveling time | 0.306 | 0.427 | 0.409 | 0.949 | 0.259 | 0.292 | 0.741 |
| Visit duration | 0.000 *** | 0.009 ** | 0.000 *** | 0.084 | 0.000 *** | 0.113 | 0.001 ** |
| | Barriers to Visiting Parks (N = 53) | | | | | | |
| **Visitor Groups** | **Lack of Time** | **Prefer Other Places** | **Poor Quality of Park** | **Long Walking Time** | **No Interest** | **Lack of Time** | **Prefer Other Places** |
| Age | 0.000 *** | 0.001 ** | 0.000 *** | 0.000 *** | 0.477 | Age | 0.000 *** |
| Gender | 0.304 | 0.557 | 0.390 | 0.830 | 0.454 | Gender | 0.304 |
| Education | 0.280 | 0.000 *** | 0.290 | 0.820 | 0.110 | Education | 0.280 |

\* Significant at *p* < 0.05; \*\* significant at *p* <.01; \*\*\* significant at *p* < 0.001.

For motivations for visiting parks, Online Supplementary Table S2 shows that walking/exercise, stress relief and visiting parks to gain inspiration ranked in the top three reasons for walking to the most frequently visited park in Guangzhou (27%, 18% and 18%, respectively). Furthermore, younger people (18 to 24 years old) were more motivated to use parks for relieving stress than older people. With increasing levels of education, there was a corresponding increase in the proportion of people visiting parks for stress relief and enjoying the views. Surprisingly, there were nearly five times more frequent visitors than infrequent visitors walking to parks to gain inspiration from the park. Similarly, frequent users were three times more likely (than infrequent users) to cite "Socialize with strangers" as the motivation for visiting. Differences in motivations by "visit duration" groups are identified in Table S2. Specifically, people spending less than 15 min in parks were more motivated by "walk/exercise". Respondents citing "Enjoy views" were more likely to be in the group with longer visitation times. Moreover, as the visiting time inside parks increased, there was a corresponding increase in the proportion of people motivated to visit parks for sightseeing.

In terms of barriers to walking to visit parks, Table S3 highlights that over 50 percent of visits less than once a month were caused by a lack of free time. Additionally, younger people, especially those aged from 18 to 24 years old, were more likely to consider "Preferring other places" to influence visiting frequency. At the same time, for elderly people aged over 50, "Poor quality of parks" and "Long walking time" were more critical barriers.

## 4. Discussion

### 4.1. Public Preferences and Park Usage

Along with an analytical appraisal of respondent park usage, exploration of public preferences provided user-based evidence for decision making for both strategic planning and park design. The results emphasize the priority of walking as a transportation mode to get to parks, especially for shorter trips of less than 15 min. This finding supports previous studies that also highlight that walking is the primary transportation used by regular park users close to parks [32,44]. Taking ease of access into account, respondents highlighted walking to visit parks was the easiest method compared to other methods of

transportation, regardless of the travel time. Such cases could be explained partly by busy traffic in metropolises, increasing residents' awareness of health and popularization of a low-carbon life.

Variations in visit duration inside parks can be influenced by a variety of activities in parks undertaken by users [15,32,45]. This is consistent with motivations for visiting parks for trips with different visiting times (Table S2) and demonstrates that visitors used parks for longer when motivated by enjoying the scenery and meeting with friends or family. In contrast, park users with shorter visiting times had a greater tendency for visiting with the intended motivation of "walking/exercise". Furthermore, people visiting parks more frequently tended to stay for shorter durations in parks. Existing research supports the finding that more frequent visits to UGS are associated with shorter walking times [46–48].

Additionally, longer walking times to parks were found to be associated with longer average visiting times spent in parks (Figure 5b). Therefore, in accord with Tu et al. [49], willingness to travel long distances and short travel distances was often found in visits for accompanying children [31] and visits for daily exercise, respectively.

The three most common motivations for visiting parks on foot were for "walking/exercise", to "relieve stress" and to "gain inspiration". These findings are in contrast to those found by Shen [31] that identified appreciating the scenery as the most common activity in UGSs. The findings here, however, are analogous to those from Chiesura [45] and Wong [50], who highlighted the popularity of sports activities including running.

Frequent use of parks (more than once a month) where people visited on foot was more common for those who appreciated traits of better views, cleanness and spatial proximity (19.36%, 10.78% and 10.57%). This finding is consistent with the research by Wang et al. [15], who identified the importance of park features including aesthetics for visitor satisfaction. Although numerous studies have stated that travel time (in terms of distance) is a fundamental factor of park usage [51–54], the importance of parks' features, for example, scenery and cleanliness, confirmed the findings of other studies that the improvement in UGSs' quality could offer potential visitors more opportunities for daily activities, and the effect of traveling distance would be a less important factor than the park's attractiveness for individuals [28,44]. Therefore, improving the overall quality of parks, such as their design, daily maintenance, etc., may potentially reduce possible negative effects associated with travel distance. In the field of accessibility modeling of UGS, it also hints that a single distance factor could not sensibly simulate the overall accessibility of UGS within a city.

With regard to reasons for infrequent park usage (less than once a month), the results indicate that a lack of time accounted for more than half of all visiting barriers cited by respondents. This is consistent with results from previous studies [31,32,50]. Other barriers identified included "prefer other leisure places" and "poor quality of park'. Diverse leisure activities have been increasing in urbanized areas and provide people with more opportunities for relaxation. Moreover, the importance of the quality of parks is highlighted by other studies, with the provision of parks lagging behind due to rapid urbanization [55], insufficient park infrastructure and poor sanitary conditions [32] and so on. These deficiencies in a park's quality are likely to have an impact on potential park users' concerns and park visitors' visiting frequency.

### 4.2. Differences across Population Groups

Several factors of preferences for park usage have been demonstrated to affect park visiting behavior, as discussed above. Additionally, differences in visiting preferences were also shown to vary with certain socio-demographic characteristics, including age, gender and education, which is consistent with the findings of Wang [15] and Zhan et al. [32].

Previous research has identified that due to physical limitations [56], age negatively correlates with UGS usage [57–59]. However, this study showed that older people (especially people aged from 41 to 50), were more likely to spend longer periods of time within parks than younger people. This could be for reasons that "lack of time" was considered the

main barrier for younger people (Table S3), and older visitors tend to have more leisure time. This result to some extent reflects that retired people prefer daily activities in parks [31], whilst younger people tend to be more infrequent park users [15]. Age also correlated with the majority of reasons for visiting parks. The proportion of people aged over 50 using the park for "walking/exercise" was higher than that of younger people (25- to 50-years-old). This finding is similar to that of other research [31,60]. It could explain the significant correlation between age and frequency revealed by this study, and similar findings from other studies [31,37,61] indicating increased park visitation with age. More elderly people may place a higher priority on maintaining physical health [31], and therefore may tend to use parks for undertaking modest exercise.

Males, as opposed to females, were found to be proportionally more likely to undertake longer walking trips to parks that were over 40 min. This could be related to the fact that females are more likely to have childcare responsibilities [62], with long walking distances to parks being inconvenient for those with children. There were no significant differences by gender in visiting frequency for shorter trips (less than 40 min). Whilst this is similar to findings from a case study in Guangzhou by Jim and Shan [63], other research has indicated that females use parks less due to fears for safety [62,64]. There were no differences between male and females in barriers limiting their frequency of park usage by foot (more than once a month). Males were found to be more likely to visit parks for gaining inspiration, whilst females were more likely to be motivated by enjoying scenery.

Education level was associated with both visiting frequency and visit duration. This is in contrast to previous research which found no correlation [31]. People with lower education levels were more likely to have shorter durations of park visit, particularly less than 15 min. This may be partially related to the finding of previous work that lower-educated groups tend to visit parks without companions [31]. Similar to a study by Gu et al. [25], higher-educated people tend to be more motivated by the views within parks. Furthermore, higher-educated groups were more likely to visit parks for stress relief. In contrast to research arguing that people with higher education attainment were more likely to visit parks in order to walk or undertake sport [31], this research found no significant correlation between education groups and the visiting reason of "walking/exercise". Higher-educated people, who also tend to be potentially higher earners overall, are less interested in park visits because of a broader diversity of choice of recreational modes [31,65]. Likewise, there were more low-educated people found to visit parks every day. Being busy and lacking time were the common barriers to walking to visit parks frequently, and there were no differences across either gender or education groups.

These differences in park visiting preferences across age groups, gender and education groups may highlight certain strategic considerations for park design and special users. For instance, firstly, as elderly people prefer longer visit durations within parks and mostly use parks for physical exercise, parks could seek to increase or improve elderly fitness equipment and specified sites for organized activities. Secondly, ensuring the provision of parks within 15 min walking distance and the inclusion of children's play facilities are important for women who prefer shorter trips to parks on foot and are more likely to have parental responsibility for children playing in parks.

### 4.3. Limitations and Outlook

Although the Internet is widely used by people, online questionnaires could miss potential park users not using social media, which could result in bias in understanding actual park usage by the population. This study employed weighting for reducing the influence of the composition of the respondent population, yet differences may persist. Non-users of UGS were excluded in this analysis due to the focus of the research resting with characteristics of park usage. Future research could focus on non-users' attitudes to UGS. Relatively modest samples for subgroup categories may influence the robustness and confidence in certain correlation analysis results. Whilst this research attempted to compare findings with other previous studies, it should highlight the difficulty in identifying directly

comparable output—both in the precise nature of the questions and methods/approaches used to identify them. This highlights the need for a robust body of evidence to influence and steer future policy. Additionally, further research is required to explore how socio-demographic factors affect use and visitation of green space typologies within a global context.

## 5. Conclusions

Facing the challenges of maximizing limited UGS resources effectively under continuing rapid expansion of build-up areas, it is crucial to have a comprehensive knowledge of UGS usage and preferences. It is therefore important to identify and understand differences between different social groups and cultural backgrounds. This study analyzed 2360 questionnaires from adult residents in Guangzhou to explore local public visitations of parks by foot in terms of: traveling time, visit duration, visiting frequency, motivations and barriers.

In summary, the results firstly identify walking as the easiest and most frequent transport method for people to their most visited park. With regard to visitation preferences, longer visiting times in parks usually corresponded to longer travel times and less frequent usage of parks. Park use was found to significantly differ among certain socio-demographic groups and types of visitors. Compared to women, the willingness of men to walk more than 40 min to visit their mostly visited parks was noted. Moreover, the time spent within a park (duration) tended to increase with age (with people aged 41 to 50 tending to have the longest duration). Young people aged 18 to 24 and people over 50 were found to be more likely to be motivated by "walking/exercise" than those aged 25 to 50. Furthermore, younger users were more likely to cite "stress relief" as their reason for visiting the park. Longer walking times and poor-quality parks were factors identified as barriers to older peoples' visits to parks. In contrast, younger users were more likely to cite a reference to other places (other than the park) as a barrier to use. Higher-educated people were found to disproportionately account for a smaller proportion of people preferring short visits to parks. There was a tendency for higher-educated visitors to be more motivated to use parks to relieve stress and enjoy the views within the park in contrast to those from lower-educated backgrounds.

Contributions of this research to public policy include the following two aspects. Firstly, our analysis demonstrates that local governments should consider the different users and their geographic footprint when planning the provision of UGS as certain groups may be more willing to walk further distances to visit UGS. Secondly, our survey indicates the correlation between socio-demographic factors and preferences for park use, demonstrating that planners should ensure that the design of urban parks should align with the population composition of the local area. For parks in these areas, designers can stimulate longer visit durations and more frequent use by aligning park design to user requirements and preference. For instance, as elderly populations have been proven to stay longer in parks but are more hindered by long walking distances and poor park quality, a park's potential could be improved by providing more benches, including well-designed footpaths, and incorporating special venues with activity equipment for the elderly.

The results highlight the importance of planning urban parks and UGSs, specifically considering the effect of socio-demographic characteristics, walking accessibility and visitation preferences on park usage. Furthermore, this study provides empirical evidence supporting the need to enhance park accessibility models considering variations in park visitation for different socio-demographic groups, for example, by providing more accurate data for travel distance by age rather than models which currently employ a single numeric value for all socio-demographic groups.

**Supplementary Materials:** The following supporting information can be downloaded at: https: //www.mdpi.com/article/10.3390/land11081219/s1, Table S1: Summary on the statistical tests undertaken for specific analysis; Table S2: Summary on reasons to visit parks by population groups; Table S3: Summary on barriers of visiting parks infrequently (less than once a month) by population groups.

**Author Contributions:** Conceptualization, Y.M., P.B. and E.L.; data collection and formal analysis Y.M.; writing—original draft preparation, Y.M. and P.B.; writing—review and editing, Y.M., P.B. and E.L.; supervision, P.B. and E.L. All authors have read and agreed to the published version of the manuscript.

**Funding:** This research received no external funding. It is associated with the Adaptive Urban Transformation project (No. EP/R024979/1), funded by the Newton Fund and Engineering and Physical Sciences Research Council (EPSRC).

**Institutional Review Board Statement:** The survey in this study was approved by university research ethics application form 030437, the Ethical Review Board of University of Sheffield (https: //www.sheffield.ac.uk/research-services/ethics-integrity/policy/approval, accessed on 28 June 2022).

**Informed Consent Statement:** Informed consent was obtained from all subjects involved in the study.

**Data Availability Statement:** Publicly available datasets were analyzed in this study and accessed on 20 October 2020. These data can be found here: (https://ghsl.jrc.ec.europa.eu/download.php? ds=pop; https://www.openstreetmap.org/).

**Conflicts of Interest:** The authors declare no conflict of interest.

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
