# Peer review of "The Influence of Socio-Demographic Factors on Preference and Park Usage in Guangzhou, China"

_land, doi:10.3390/land11081219_

Round 1

Reviewer 1 Report

My comments to the authors:

The authors did very good work in the analyses and presentation of their results. The weakness, as the authors already stated in the Line 345-350 of their manuscript, is also notable.

One suggestion to the authors is to remove the reference #33.

As stated in that reference paper, “(t)he case study was conducted in a botanic garden in Central Florida.” The contents and quality of this type of ‘botanic garden’ is significant different from the type of ‘urban green space (UGS)’ in a very large and heavily populated city like Guangzhou. Thus, visitors experience in such a garden is different, and shouldn’t be the same with visitors’ experiences in a UGS near by where they live in large cities.

The authors of this manuscript only refer this reference in one place at the Line 69 to 70, as:

“Additionally, the attractiveness of parks could be reflected by an individual's visiting behavior [33], ….”.

Clearly, the “parks” referred here are different from the “botanic garden”, studied in that reference paper #33. This is also less relevant to the point of UGS, which the authors wanted to state here in this section.

Thus, it’s better to remove this reference.

Similarly, to the reference #34, I’d suggest the authors to pay attention to the differences between a ‘national park’ and a ‘UGS’, in term of visitors’ experience.

I suggestion the authors to add some sentences in the Introduction section, state the similarity and differences among a national park, a botanic garden, a park, and a UGS in a large city, etc. Understanding better on these will be very useful for such research topic and issues as the authors did.

Reviewer 2 Report

The paper presents a study about the influence of the socio-demographic factors on the preference and usage of the Urban Green Space (UGS). There are some issues that the authors should better explain/present:

-          Abstract is too long and information about data processing and the policies implications for the main results is missing.

-          Introduction – it provides a detailed presentation of the current state of the art. To improve the overall section the authors should better emphasise any existing typologies between concerning the influence of different socio-demographic factors in different regions of the world.

-          Material and methods – the methods for data analysis should be better explain. Also, the number of responses (2,360) must be explained using statistical methods.

-          The implications of the results for the public policy makers must be better emphasised.
